# RaScALL: Rapid (Ra) screening (Sc) of RNA-seq data for prognostically significant genomic alterations in acute lymphoblastic leukaemia (ALL)

Jacqueline Rehn[1,2], Chelsea Mayoh[3,4], Susan L Heatley[1,2,5], Barbara J McClure[1,2], Laura N Eadie[1,2], Caitlin Schutz[1], David T Yeung[1,2,6], Mark J Cowley[3,4], James Breen[7,8‡]*, Deborah L White[1,2,5,9,10‡]

1 Blood Cancer Program, Precision Cancer Medicine Theme, South Australian Health & Medical Research Institute (SAHMRI), Adelaide, South Australia, Australia, 2 Faculty of Health and Medical Science, University of Adelaide, Adelaide, South Australia, Australia, 3 Children's Cancer Institute, Kensington, New South Wales, Australia, 4 School of Clinical Medicine, UNSW Sydney, Sydney, New South Wales, Australia, 5 Australian and New Zealand Children's Oncology Group (ANZCHOG), Clayton, Victoria, Australia, 6 Department of Haematology, Royal Adelaide Hospital and SA Pathology, Adelaide, South Australia, Australia, 7 Black Ochre Data Labs, Telethon Kids Institute, Adelaide, South Australia, Australia, 8 Australian National University, Canberra, Australian Capital Territory, Australia, 9 Australian Genomics Health Alliance (AGHA), Parkville, Victoria, Australia, 10 Faculty of Sciences, University of Adelaide, Adelaide, South Australia, Australia

‡ These authors are joint senior authors on this work.
* Jimmy.Breen@telethonkids.org.au

**Data Availability Statement:** Raw FASTQ files for ALL samples in the study cohort are available from Genome-phenome Archive (EGAS00001006460).

## Abstract

RNA-sequencing (RNA-seq) efforts in acute lymphoblastic leukaemia (ALL) have identified numerous prognostically significant genomic alterations which can guide diagnostic risk stratification and treatment choices when detected early. However, integrating RNA-seq in a clinical setting requires rapid detection and accurate reporting of clinically relevant alterations. Here we present RaScALL, an implementation of the k-mer based variant detection tool *km*, capable of identifying more than 100 prognostically significant lesions observed in ALL, including gene fusions, single nucleotide variants and focal gene deletions. We compared genomic alterations detected by RaScALL and those reported by alignment-based *de novo* variant detection tools in a study cohort of 180 Australian patient samples. Results were validated using 100 patient samples from a published North American cohort. RaScALL demonstrated a high degree of accuracy for reporting subtype defining genomic alterations. Gene fusions, including difficult to detect fusions involving *EPOR* and *DUX4*, were accurately identified in 98% of reported cases in the study cohort (n = 164) and 95% of samples (n = 63) in the validation cohort. Pathogenic sequence variants were correctly identified in 75% of tested samples, including all cases involving subtype defining variants *PAX5* p.P80R (n = 12) and *IKZF1* p.N159Y (n = 4). Intragenic *IKZF1* deletions resulting in aberrant transcript isoforms were also detectable with 98% accuracy. Importantly, the median analysis time for detection of all targeted alterations averaged 22 minutes per sample, significantly shorter than standard alignment-based approaches. The application of RaScALL

BAM files for the validation cohort containing 100 ALL samples are available from the European Genome-phenome Archive (accession numbers EGAD00001004461 and EGAD00001004463). Target sequences described in this study can be accessed via figshare (https://doi.org/10.25909/19372559.v1). Code and additional scripts for running RaScALL are available from the GitHub repository (https://github.com/j-rehn/RaScALL).

**Funding:** This work was supported by funding from the Australian Genomics Health Alliance (AGHA) (1113531 to DLW), Beat Cancer (DLW, www.cancersa.org.au), and the Leukaemia Foundation (DLW, www.leukaemia.org.au). DLW is an NHMRC R.D. Wright II Fellow (1160833) and a Beat Cancer Principal Research Fellow. DTY is an NHMRC Early Career Fellow (1146253). SLH is The Kids' Cancer Project Fellow (www.thekidscancerproject.org.au). LNE is the Peter Nelson Leukaemia Research Fellow (www.cancersa.org.au). JR and CM were supported by funding from the Australian Research Training Program (RTP). The funders had no role in study design, data collection and analysis, decision to publish, or preparation of the manuscript.

**Competing interests:** The authors have declared that no competing interests exist.

enables rapid identification and reporting of previously identified genomic alterations of known clinical relevance.

## Author summary

Acute lymphoblastic leukaemia is a life-threatening malignancy characterised by various genomic alterations. RNA-sequencing (RNA-seq) is an effective method for identifying genomic lesions that drive patient disease, providing critical information about patient prognosis and treatment. However, analysis of RNA-seq data to identify clinically relevant genomic alterations is often time and resource intensive. Here, we present RaScALL, an integrated platform for rapid (Ra) screening (Sc) of RNA-seq data to identify genomic alterations of clinical significance in acute lymphoblastic leukaemia (ALL). RaScALL can detect more than 100 lesions of clinical importance from raw RNA-seq data, including gene fusions and sequence variants. A comparison of alterations detected by RaScALL and those reported by *de novo* variant detection tools confirmed that RaScALL is highly accurate and computationally efficient, with an average runtime of fewer than 30 minutes. Where RNA-seq data is available RaScALL provides rapid detection and reporting of prognostically significant lesions, which may ultimately assist with patient risk-stratification and therapeutic triage.

## Introduction

Acute lymphoblastic leukaemia (ALL) is a life-threatening haematological malignancy characterised by a wide variety of genomic alterations. It is the most common childhood cancer and accounts for 15–25% of adult acute leukaemia [1]. Although the introduction of risk-adapted therapy has significantly improved treatment outcomes, particularly in paediatric cohorts, relapsed and refractory disease remains lethal. High-risk genomic lesions are disproportionately frequent in adolescent and adult patients, contributing to higher relapse rates and poorer overall survival [2]. Timely and accurate identification of genomic variants involved in disease pathogenesis is therefore essential both for patient risk stratification and the identification of individuals who may benefit from precision medicine approaches [3].

In the past decade, next-generation sequencing (NGS) of large patient cohorts has greatly expanded our understanding of the genomic alterations driving leukaemia progression and identified additional molecular targets amenable to targeted therapy [4–6]. This includes the identification of a variety of gene fusions, many of which are cytogenetically cryptic [3,7]. These gene fusions often result in chimeric transcripts consisting of exons from two different genes, with resultant chimeric proteins driving leukemogenesis [2]. Several of these fusions result in the activation of kinase signalling, some of which may be abrogated by currently available therapies [8,9]. Alternatively, a gene fusion can result in altered expression of cytokine receptors [10–12] or transcription factors [13,14] due to the placement of these genes under alternate transcriptional regulation [12–14]. Each of these genomic alterations is associated with distinct gene expression and risk profiles [4–6].

Missense mutations resulting in loss of function of key lymphoid transcription factors have also been detected and represent gene distinct expression and risk profiles [4,5]. Furthermore, studies have revealed several cooperating mutations that are either associated with relapse or indicative of high-risk disease. For example, single nucleotide variants (SNVs) affecting genes

of the RAS pathway have been associated with relapsed ALL [15,16]. Activating mutations of *JAK2* and *CRLF2* occurring concurrently in patients with overexpression of CRLF2 promote JAK-STAT activation and high-risk disease [10,17,18]. Focal deletions of the lymphoid transcription factor IKAROS *(IKZF1)* and other transcription factors critical to B-cell development are independent prognostic factors indicative of poor outcomes [10,19–21]. Early detection of cooperating mutations indicative of high-risk disease is likewise essential for the identification of patients who may benefit from intensified therapy.

These advances have led to an increase in NGS, particularly RNA-sequencing (RNA-seq), for identifying genomic lesions indicative of disease risk. However, full implementation of NGS in a clinical setting depends upon the rapid and accurate identification of clinically relevant variants [7,22]. Currently, there is no standard, 'Best Practice' RNA-seq diagnostic pipeline for the detection of genomic variants [23], with independent research groups developing their own in-house bioinformatic pipelines for analysis [4–6,24]. Multiple programs exist for *de novo* detection of sequence variants and gene fusions from RNA-seq data but vary in their sensitivity to detect clinically relevant lesions [23,25–27]. Furthermore, these programs report large numbers of expressed variants or chimeric transcripts, many of which are either false positives or do not represent clinically relevant alterations. Consequently, extensive knowledge-based filtering and manual curation are required to identify alterations for clinical reporting [25].

Recently, a new command line tool for targeted detection of genomic variants was developed, which relies on a k-mer based or pseudo-alignment approach rather than the direct mapping of reads to a reference genome. This application, *km* [28], differs from standard analysis by limiting the search space to a set of target sequences representing known genomic alterations. Raw RNA-seq data, decomposed into substrings or k-mers, is assembled across target sequences by an algorithm that identifies diverging paths indicative of substitutions, insertions, or deletions. Pseudo-alignment is incredibly memory and computationally efficient, leading to a considerable reduction in analysis time [28]. However, the application of *km* to ALL patient samples requires the development of target sequences representing disease-specific lesions and an evaluation of performance against more commonly employed *de novo* variant detection tools.

Here we present RaScALL, an integrated pipeline utilising *km* for rapid (Ra) screening (Sc) of RNA-seq data for clinically relevant variants in patients with ALL. RaScALL contains target sequences for the detection of more than 100 clinically relevant genomic alterations, including 75 distinct gene fusions, with the functionality to generate custom target sequences to identify novel variants. We demonstrate the ability of RaScALL to detect subtype defining genomic alterations rapidly and accurately, including gene fusions involving the *IGH* locus that are frequently under-reported by alignment-based fusion calling tools. We also successfully identify cooperating mutations, including single nucleotide variants (SNVs) and intragenic *IKZF1* deletions, contributing to patient prognosis.

## Results

### Study cohort overview

To investigate the applicability of targeted methods for detecting disease-defining genomic alterations a study cohort consisting of 180 patient samples representing common prognostically significant genomic lesions was employed. A separate validation cohort of 100 B-ALL samples representing a broader range of B-ALL subtypes was utilised to verify these results. The study cohort including B-ALL (n = 160), T-ALL (n = 18), acute undifferentiated leukaemia (n = 1) and acute bi-phenotypic leukaemia (n = 1) was selected, from a larger pool of

samples referred to the South Australian Health and Medical Research Institute for genomic analysis (S1 Table). Informed consents were given under HREC approved clinical trial protocols. For this study cohort, subtype defining gene fusions and other recurrently observed genomic alterations were initially identified by a *de novo* variant calling and fusion detection pipeline. A 'Best Practice' GATK RNA-seq variant calling workflow [29] was applied to raw patient RNA-seq datasets, relying on haplotype caller for identification of SNVs and Insertion/ Deletions (InDels).

Gene fusion detection was performed using four fusion calling algorithms: *FusionCatcher* [30], *SOAPfuse* [31], *JAFFA* [32] and *arriba* [23]. Apart from *IGH* rearrangements (*IGH*r), consensus between two or more algorithms was required for reporting clinically relevant gene fusions. Rearrangements of *IGH* are frequently under-reported by many fusion calling algorithms due to poor alignment across the *IGH* region of the genome [22,25], making consensus between callers more difficult in these instances. Fluorescence *in situ* hybridisation (FISH) was therefore utilised to confirm *IGH-CRLF2* gene fusions (S1 Table) while *IGH-EPOR* alterations were verified by visual inspection of the BAM file. Patients with *IGH-DUX4* rearrangements were identified via gene expression profiling (S1 Fig). All patient samples were sequenced to high coverage (mean 70M PE reads). A minimal set of three samples (*ETV6-ABL1*, n = 2) representing each subtype defining gene fusion or SNV was included for testing and development of RaScALL, with additional samples selected to maximise common genetic subtypes along with *IGH-CRLF2* and *IGH-DUX4* alterations that are difficult to identify by alignment-based tools. The reported genomic alteration, along with age and subtype classification, is represented in Fig 1.

## Overview of RaScALL

RaScALL is an implementation of *km* for rapid screening of raw RNA-seq data to identify a predefined list of clinically relevant genomic alterations in ALL. Variant detection requires target sequences representing known fusion breakpoints or regions surrounding an SNV or InDel of interest. Target sequences of 62b length were generated for 25 gene fusions involving exon-exon breakpoint sequences affecting patients in the study cohort (S2 Table). For *IGH-E-POR* and *IGH-CRLF2* gene fusions, with highly variable breakpoint locations, targets represented sequence upstream of commonly reported *CRLF2/EPOR* breakpoints combined with sequence from all *IGK*, *IGHJ* and *IGHD* alleles. Rearrangements of *DUX4* (*DUX4*r) likewise have highly variable breakpoint locations but result in high-level expression of *DUX4*, which is otherwise absent in leukaemic and non-leukaemic lymphocytes. Thus, targets representing the *DUX4* mRNA transcript were created for the detection of *DUX4* expression as an indicator of *DUX4*r (see Materials and Methods for details). Additional targets consisting of reference sequences surrounding 16 clinically relevant SNVs were also generated (S2 Table).

To perform variant detection with RaScALL, raw FASTQ files from patient samples are indexed by *jellyfish* v2.2.6 [33] to generate a k-mer count table (k = 31) and passed to *km* [28] for assembly of k-mers across the provided target sequences. The output is then filtered, collated, and annotated using a meta-caller R script to generate a final output file containing reportable genomic alterations. Only instances where each k-mer in the target sequence was identified were considered positive identification of the fusion or sequence variant. A command line utility to generate custom targets for gene fusions, SNVs and InDels is also provided (Fig 2A).

## Memory and run-time evaluation

Speed and computational resource requirements are important considerations for bioinformatic analysis in a clinical setting. Variant detection with RaScALL is a two-step process.

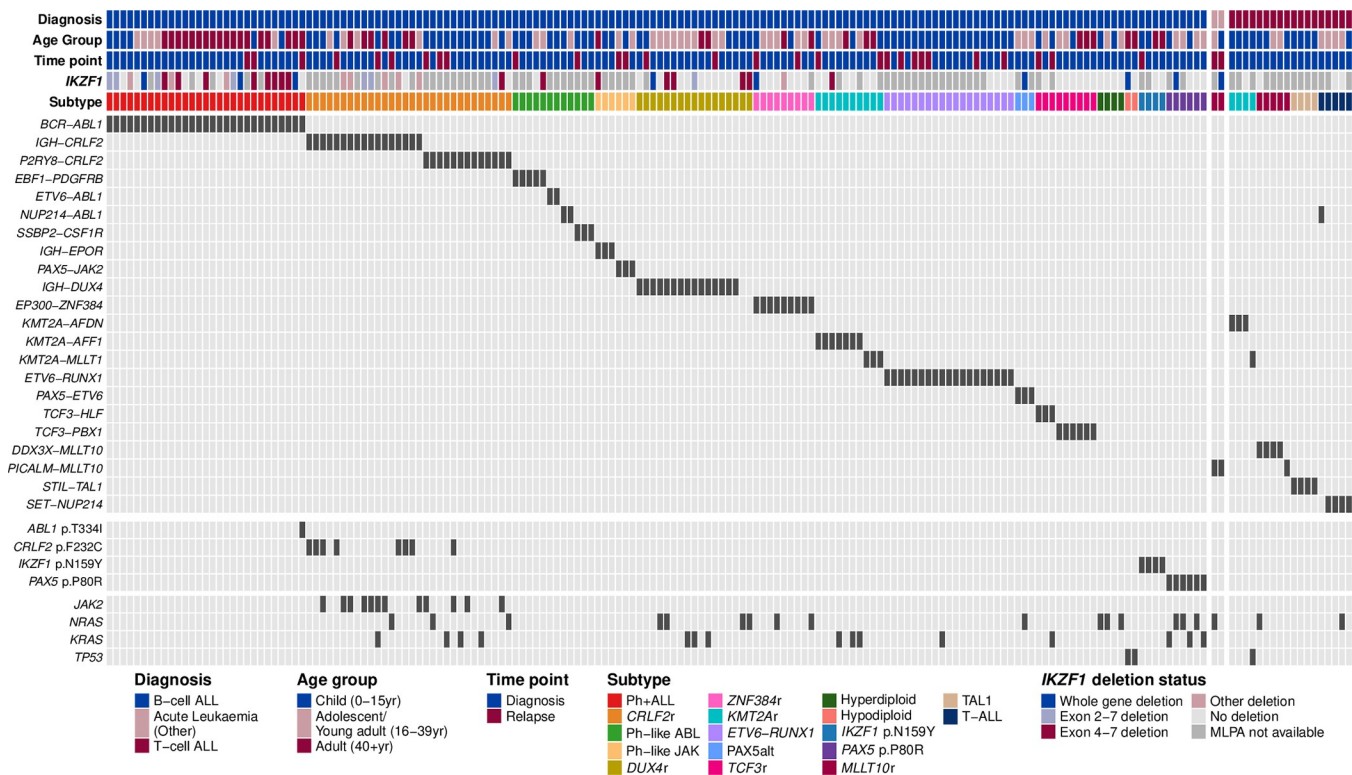

**Fig 1. Genomic alterations for patients within the study cohort.** Reported genomic alterations for 180 ALL samples analysed by RNA-seq, including 160 B-ALL patient samples (left panel), 18 patients samples classified as T-ALL (right panel), and two samples with acute undifferentiated leukaemia or acute bi-phenotypic leukaemia (middle panel). Each column represents a single patient. Patients were classified into a recognised subtype based on the detected driver mutation. For each patient, the deletion status of *IKZF1*, as reported by multiplex ligation-dependent probe amplification, as well as patient diagnosis, age group and sample timepoint are shown and coloured according to the key. Clinically relevant gene fusions detected with RNA-seq are indicated in the middle panel and shown in dark grey. The bottom panel indicates samples for which known pathogenic SNVs were detected by GATK Haplotype caller (shown in dark grey).

RNA-seq data is initially indexed with the command line tool *jellyfish* [33] to generate a k-mer count table for the patient. *Km* [28] then utilises the count table to assemble the target sequences from overlapping k-mers and identify divergent paths representing variants. Both steps contribute to the overall memory requirements and analysis time and were evaluated to assess the effects of sequencing length and depth on runtime. Maximum memory requirements for indexing were dependent on the size of the RNA-seq dataset, read length and sequencing depth, consistent with previous benchmarking studies [34]. Memory requirements increased from 4.7GB for datasets with 75b length PE reads and depth of 40 million reads up to 17.8G for datasets involving 150bp length and 80 million reads (S3 Table). Runtime likewise increased with the size of the dataset to process but could be minimised by increasing the number of available threads. When performed with 12 threads, on data with 75b read length and read depth of 80M, the mean runtime was 6.5 minutes (S3 Table).

Memory requirements and runtimes for variant detection with *km* are much smaller than the indexing step, although do increase with the increasing size of the RNA-seq data (Table 1). Runtime for detection of *IGH-CRLF2 and IGH-EPOR* gene fusions was considerably longer than detection of other genomic alterations, averaging 8.7 minutes for datasets of 50-70M reads, compared with less than 30 seconds for all other targets. Targets for detection of *IGH-CRLF2* and *IGH-EPOR* represent sequences from regions placed in proximity when a rearrangement occurs rather than the exact sequence generated by fusions producing exon-

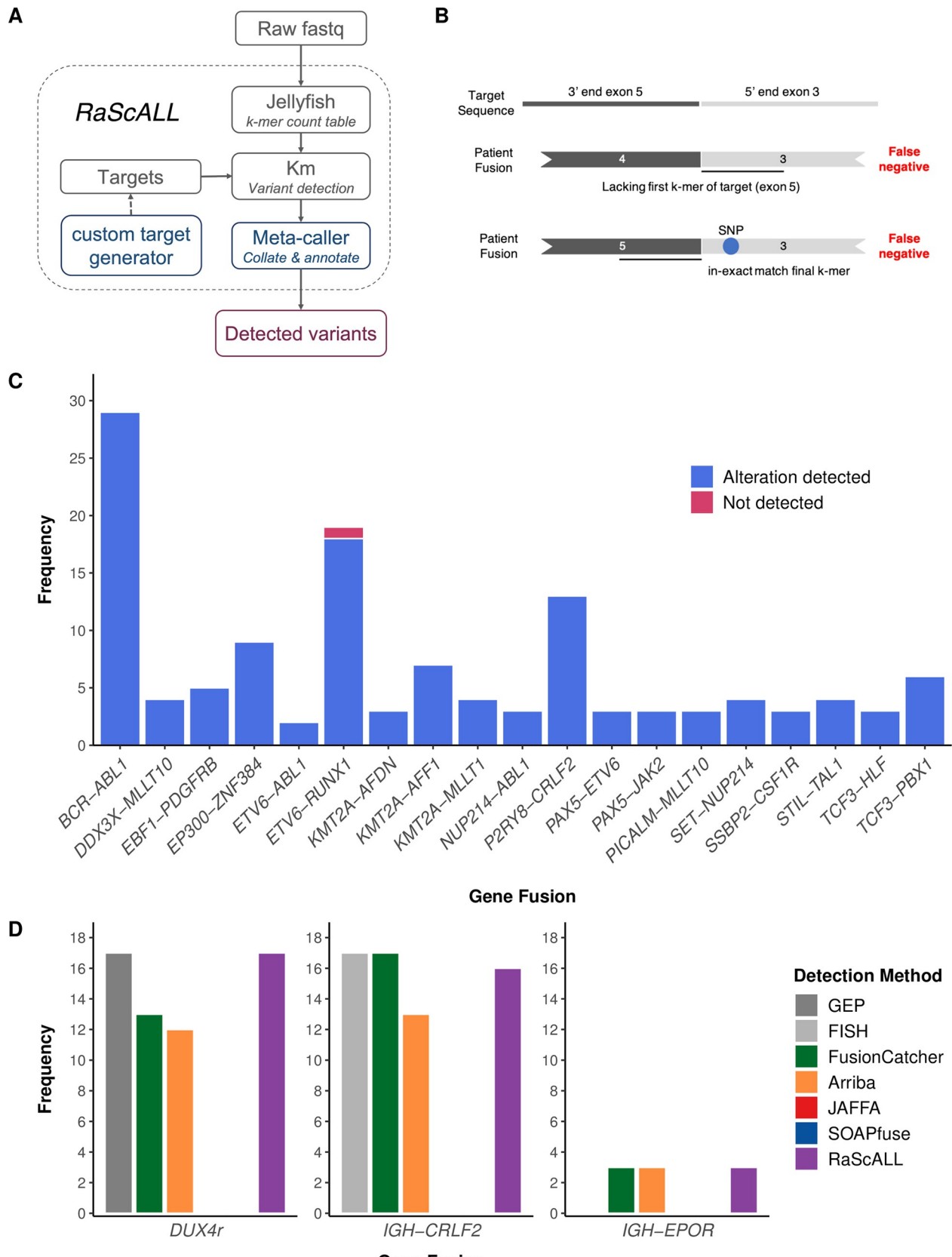

**Fig 2. Detection of subtype-defining gene fusions by RaScALL.** (A) RaScALL workflow. RaScALL is a self-contained command line utility for detecting ALL specific variants. RNA-seq data in the form of raw FASTQ files are indexed by *jellyfish* to generate a k-mer count table. Next, gene fusions, focal gene deletions, SNVs and InDels are detected with *km* for genomic regions specified in the target set and the output is filtered to provide a single file with clinically relevant alterations for reporting. (B) Schematic representation of gene fusion target sequences (top panel) and causes of false-negative reports. A false negative occur when the patient possesses an alternate breakpoint lacking one of the exons in the gene fusion target (middle panel), or due to the presence of a SNP preventing an exact match of the first or final k-mer in the target sequence (bottom panel). (C) Frequency of detection of non-*IGH* gene fusions by RaScALL in the study cohort. Blue bar indicating that RaScALL detected the same gene fusion reported by two or more alignment-based fusion calling algorithms and a single sample in pink for which RaScALL failed to detect a clinically reported gene fusion. (D) Frequency of *IGH-CRLF2*, *IGH-EPOR* or *DUX4*r in the study cohort as determined by FISH for *IGH-CRLF2* or gene expression profiling (GEP) for *DUX4*r (grey). Coloured bars indicate the number of samples for which RaScALL (purple) or one of 4 alignment-based fusion calling algorithms detected the rearrangement in the given patient sample.

exon breakpoints. The large diversity of sequence and high-level expression associated with the *IGH* locus can account for this additional analysis time. Overall analysis time for sample indexing and assessment of targeted variants for the study cohort averaged 16.8 minutes. Based on a benchmarking assessment utilising 9 samples, only *arriba* demonstrated comparable speed (mean 13.5 minutes) with run times increasing to 14.5 hours for *SOAPfuse* (S2 Fig). Importantly, the speed of *arriba* was dependent on significantly higher memory utilisation than RaScALL (mean 34.4GB vs 10.8GB). While we recognise that this is not a fair comparison given that RaScALL can only detect a limited repertoire of gene fusions compared to arriba, the high speed and lower memory utilisation represent a significant saving in bioinformatic costs in a cloud computing environment where pricing is dependent on both variables [35].

## RaScALL accurately detects subtype defining gene fusions

RaScALL was applied to samples in the study cohort and alterations detected by RaScALL were then compared to the reported gene fusions for the patient based on the *de novo* bioinformatic analysis using a standard reference genome alignment approach (S4 Table). There were 127 non-*IGH*r subtype defining gene fusions, of which 126 were detected with RaScALL (Fig 2C). The reciprocal gene fusion (*ABL1-BCR*, *AFF1-KMT2A*, *MLLT10-PICALM*, *MLLT10-DDX3X* and *ZNF384-EP300*) was also identified for 49 of 50 patient samples (S4 Table). The false negative result, in which RaScALL failed to identify a non-*IGH*r gene fusion reported by all alignment-based fusion calling algorithms, was for a patient possessing an *ETV6-RUNX1* translocation with a novel breakpoint lacking the first k-mer of the target sequence provided (Fig 2B middle panel). The generation of an additional target with this uncommon breakpoint resulted in the correct identification of the fusion. No fusions were reported for the 16 cases in the study cohort identified as fusion negative using the *de novo* variant detection pipelines (S4 Table).

Clinically relevant *IGH*r are detected less reliably from RNA-seq than other gene fusions [23,25] and were thus considered separately. *IGH-CRLF2* was detected by RaScALL in 16 of 17

**Table 1. Memory and runtime requirements for variant detection from previously indexed RNA-seq data.** Values represent mean time and memory for samples of specified sequencing depth.

| Target set | No. target sequences | Target length | *Km* run time (s) | | | Peak RSS (GB) | | |
|---|---|---|---|---|---|---|---|---|
| | | | Low (<50M) | Medium (50-80M) | High (80+M) | Low (<50M) | Medium (50-80M) | High (80+M) |
| SNV | 14 | 69–72 | 0.15 | 0.15 | 0.17 | 0.04 | 0.05 | 0.05 |
| DUX4 | 12 | 70 | 0.08 | 0.15 | 0.18 | 0.03 | 0.05 | 0.05 |
| *IKZF1* deletion | 3 | 62 | **1.67** | 2.71 | 2.84 | 0.10 | 0.13 | 0.12 |
| Fusion | 67 | 62 | 4.73 | 5.09 | 6.70 | 0.90 | 0.76 | 0.88 |
| *IGH-EPOR* | 1656 | 64–68 | 13.88 | 27.57 | 32.91 | 0.87 | 0.98 | 1.08 |
| *IGH-CRLF2* | 525 | 81–100 | 400.26 | 342.92 | 505.12 | 2.66 | 1.08 | 3.57 |

cases for which the translocation had been identified by FISH with no false positive results (Fig 2D). For the remaining sample, the *IGH-CRLF2* fusion breakpoint occurred outside the range of targets generated. *IGH-EPOR* was detected in all 3 samples carrying this genomic alteration. Multiple targets representing the *DUX4* mRNA transcript were detected in all 17 samples previously identified as *DUX4*r by gene expression profiling, including two samples where a *DUX4* fusion was not identified by any alignment-based fusion calling algorithm (Fig 2D). Mean minimum k-mer coverage across targets for these samples ranged from 76–2690 k-mers with all samples indicating the presence of 6–9 DUX4 targets. Additionally, there were 8 non-*DUX4*r samples for which 1 or 2 targets could be assembled, but all reported mean minimum coverage of 2–4 kmers (S3 Fig). Application of filters to exclude calls where fewer than 4 DUX4 targets were detected with a minimum of 10 k-mers effectively excluded all potential false-positive results. Overall, detection rate of clinically relevant gene fusions by RaScALL was comparable with the most sensitive alignment-based caller FusionCatcher (S4 Fig).

## Detection of *IKZF1* deletions

Deletion of *IKZF1*, a transcription factor required for lymphoid development, has been identified as an independent prognostic indicator of high-risk disease [20,21]. Two commonly observed intragenic *IKZF1* deletions, exon 2–7 and exon 4–7, result in aberrantly spliced transcripts of *IKZF1* leading to haploinsufficiency of the transcription factor or production of a dominant-negative isoform ik6 [36,37]. Deletions in *IKZF1* are typically identified through DNA sequencing techniques, single nucleotide polymorphism microarrays (SNParray) [36], or through low-throughput targeted detection approaches such as multiplex ligation-dependent probe amplification (MLPA) [38,39]. Intragenic deletions resulting in aberrantly spliced transcripts can however be detected from RNA-seq data [40].

Within the study cohort, *IKZF1* deletions were identified by RaScALL using targets representing aberrantly spliced transcripts. Of the 98 samples with available MLPA data, transcripts supporting an exon 4–7 or exon 2–7 *IKZF1* deletion were correctly detected in 27 patient samples. A target seqeunce for the detection of exon 2–3 deletions was also generated, but this deletion was not present in any of the samples analysed by MLPA (S4 Table). There were 9 discordant results in which RaScALL indicated the presence of transcripts representing an *IKZF1* deletion with limited supporting k-mers that were not supported by MLPA (Fig 3A). These likely represent low numbers of aberrantly spliced *IKZF1* transcripts and not true deletions. Application of a filter to exclude *IKZF1* deletion predictions with a minimum k-mer coverage of less than 10, resulted in the condordant detection of 24 *IKZF1* deletions and one discordant exon 4–7 *IKZF1* deletion, occurring in a Ph+ALL hyperdiploid patient. Additional *IKZF1* deletions, including whole gene deletions and deletions of exons 4–8 went undetected (Fig 3B and S4 Table). Deletions of *IKZF1* were detected by RaScALL in an additional 25 samples lacking MLPA (S4 Table). While these deletions could not be verified, all occurred in patient samples representing subtypes commonly associated with *IKZF1* deletion.

## Detection of missense mutations

While gene fusions define the majority of ALL subtypes, loss of function sequence variants *PAX5* p.P80R and *IKZF1* p.N159Y have been reported as independent driving events in leukaemic development, representing distinct ALL subtypes [5,41]. Furthermore, cooperating missense mutations are often necessary for leukaemic transformation or are associated with high-risk disease. For example, activating mutations of *JAK2* are frequently observed with rearrangements of *CRLF2* and associated with the high-risk Ph-like subtype [10,17], while *TP53* variants are common in hypodiploid ALL [42,43]. Given the prognostic significance of

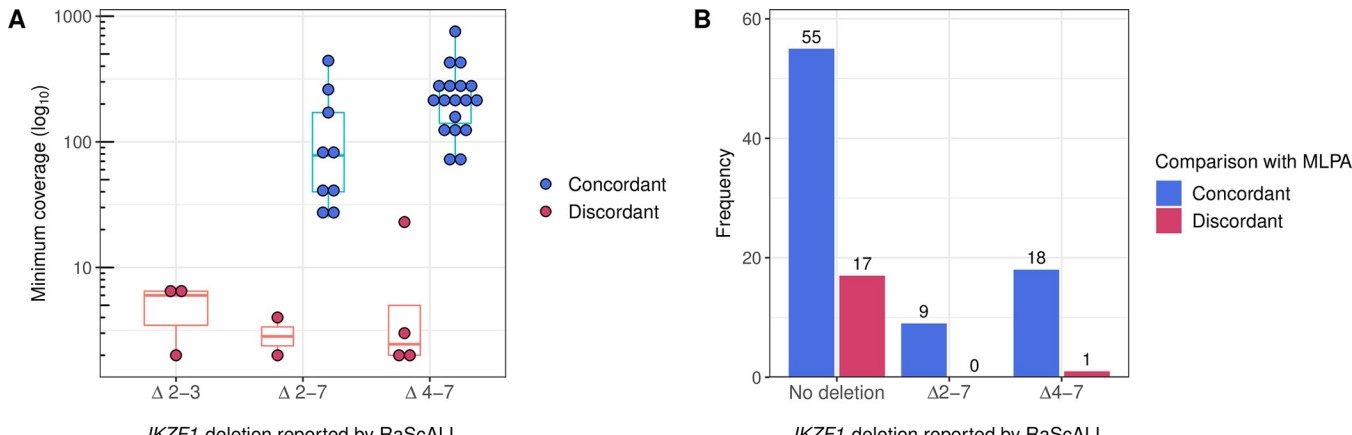

**Fig 3. Detection of *IKZF1* intragenic deletions with RaScALL.** Samples were analysed by RaScALL for the presence of target sequences representing *IKZF1* Δ2–3, Δ2–7 and Δ4–7 deletions. Deletion events reported by RaScALL were then compared to *IKZF1* deletions reported by MLPA analysis using P335 and P202 kits. (A) Minimum coverage (k-mers) across target sequences (log₁₀ scale) for targets representing *IKZF1* Δ2–3 deletion, Δ2–7 deletion or Δ4–7 deletion. Points are coloured concordant (blue) when the same *IKZF1* deletion was identified by MLPA or discordant (pink) when MLPA reported no *IKZF1* deletion for the sample. (B) Sample frequency for *IKZF1* status reported as either no deletion, Δ2–3 deletion, Δ2–7 deletion or Δ4–7 *IKZF1* deletion (minimum k-mer coverage > = 10). Bars are coloured according to whether the *IKZF1* deletion reported by RaScALL was supported (Concordant, dark blue), or not supported (discordant, pink) by MLPA analysis.

these variants, target sequences were generated for a small range of recurrently observed pathogenic SNVs (Table 2). Calls made by RaScALL within the study cohort were compared with SNVs reported by GATK HaplotypeCaller.

Pathogenic SNVs were detected in 59 of 72 cases (82%), including all instances of subtype defining variants *PAX5* p.P80R (n = 6) and *IKZF1* p.N159Y (n = 4). RaScALL detected *JAK2*, *NRAS* and *KRAS* variants at a lower frequency than was reported by GATK (Table 2 and Fig 1). Failure to identify variants in these three genes was attributed to low read coverage or low variant allele frequencies, resulting in fewer than 5 k-mers supporting the variant across each position of the target sequence. A variant occurring in *ABL1* went undetected by RaScALL despite high coverage across the genomic region. Inspection of the BAM file revealed an

**Table 2. Single nucleotide variants targeted by RaScALL.**

| Gene | Amino acid substitution (p.) | Detection rate | Clinical relevance | Reference |
|---|---|---|---|---|
| *PAX5* | P80R | 6/6 (100%) | Loss of function | [41] |
| *IKZF1* | N159Y | 4/4 (100%) | Perturbed *IKZF1* function | [5] |
| *ABL1* | T315I | 0/1 (0%) | Resistance to tyrosine kinase inhibitors | [9] |
| *CRLF2* | F232C | 8/8 (100%) | Gain of function mutation | [44] |
| *JAK2* | I682F | 0/1 (0%) | Gain of function mutation | [45] |
| | R683G, R683S, R683T | 4/8 (50%) | Gain of function mutation | [45] |
| | T875N | 2/2 (100%) | Gain of function mutation | [46] |
| | R938Q | 1/1 (100%) | Resistance to *JAK2* inhibitors | [47] |
| *NRAS* | G12A, G12C, G12D, G12S, G12V | 11/13 (85%) | Associated with inferior outcome and increased risk of relapse | [16,48] |
| | G13D | 2/4 (50%) | | |
| | Q61H, Q61K, Q61P | 4/4 (100%) | | |
| *KRAS* | G12C, G12D, G12S, G12V | 11/12 (92%) | Associated with inferior outcome and increased risk of relapse | [16,48] |
| | G13D | 3/3 (100%) | | |
| | A146P, A146V | 0/2 (0%) | | |
| *TP53* | R209Q/R248Q, R209W/R248W | 3/3 (100%) | Associated with poor prognosis | [43,49] |

additional SNV on the same allele upstream of the targeted amino acid position. An in-exact match to the final k-mer of the provided target sequence resulted in false-negative detection of these two pathogenic variants (Fig 2B).

## RaScALL accurately identifies subtype defining alterations in a validation cohort

Previously described gene fusions and SNVs represent only a portion of clinically relevant genomic alterations affecting ALL patients, limiting the applicability of RaScALL. To enable detection of a broader range of genomic alterations, we performed a literature search identifying an additional 46 gene fusions and 26 SNVs reported in multiple ALL patients and associated with functional effects [5,8,50–67]. Targets for these variants, and any alterations undetected in the study cohort, were generated using RaScALL's custom target generation tool (see Materials and Methods), resulting in an expanded target set representing 75 distinct gene fusions and sequence variants affecting 14 cancer-related genes (S5 Table). Run time for creation of the additional 1108 target sequences by the custom target generation tool was less than 3 minutes. Targets representing intragenic deletions of *ERG* and *PAX5* were also manually generated. To assess applicability against a greater number of B-ALL subgroups than was represented by the study cohort, a random selection of 100 B-ALL patient samples from a published study by Gu *et al.* 2019 [5] were analysed by RaScALL using this updated target set (target set B). This validation cohort contained samples representing B-ALL subgroups absent from the study cohort including *MEF2D*r and *PAX5*alt (S6 Table). The effectiveness of RaScALL to identify subtype defining genomic alterations and clinically relevant cooperating variants was determined by comparing the genomic alteration identified by RaScALL with those reported by Gu *et al.* 2019 [5] in the original publication for these selected samples.

RaScALL reported concordant gene fusions in 63 of 66 patient samples (Fig 4A) and identified the reciprocal gene fusion in 21 samples (S5 Table). No fusion was reported for the 29 samples lacking a clinically relevant gene fusion. Discordant results, based on alterations reported in Gu *et al.* 2019 [5], were observed in 7 patient samples, where RaScALL either failed to detect the reported fusion (n = 3) or falsely predicted a gene fusion (n = 4). Undetected fusions included one instance of *IGH-CRLF2*, one sample with *BCR-ABL1* with limited read coverage over the fusion breakpoint, and one sample with *KMT2A-AFF1* with a breakpoint not represented in the target set, although the reciprocal fusion (*AFF1-KMT2A*) was identified by RaScALL. False-positive gene fusions including *P2RY8-CRLF2* (n = 3) and *DDX3X-MLLT10* (n = 1) were detected with minimal coverage over the target sequence (minimum k-mer coverage < 5).

Missense mutations and insertions were correctly identified by RaScALL in 75% of cases (43 of 57 reported sequence variants detected) (Fig 4B). As with the initial study cohort, missed variants were attributed to limited read coverage (low-level gene expression or low variant allele frequency). An *IL7R* p.T244I variant was correctly detected in 33 patient samples but determined to represent a benign polymorphism based upon the population allele frequency reported in the Genome Aggregation Database (gnomAD), while an *IL7R* p.S252ins predicted in 19 samples represented a false positive variant attributed to a homopolymer sequencing artefact. While not reported by Gu *et al.* 2019 [5], pathogenic *CRLF2* and *TP53* variants were correctly identified in 6 patient samples and confirmed by visual inspection with IGV. Focal deletions of *IKZF1*, *PAX5* and *ERG* were likewise identified in 31 cases.

Subtype defining alterations were detected in 67 of 100 patients. In the remaining 33 samples, 23 did not have a reportable fusion or loss of function sequence variant, while 8 samples had rare gene fusions with no available target sequence. Causative lesions were undetected by

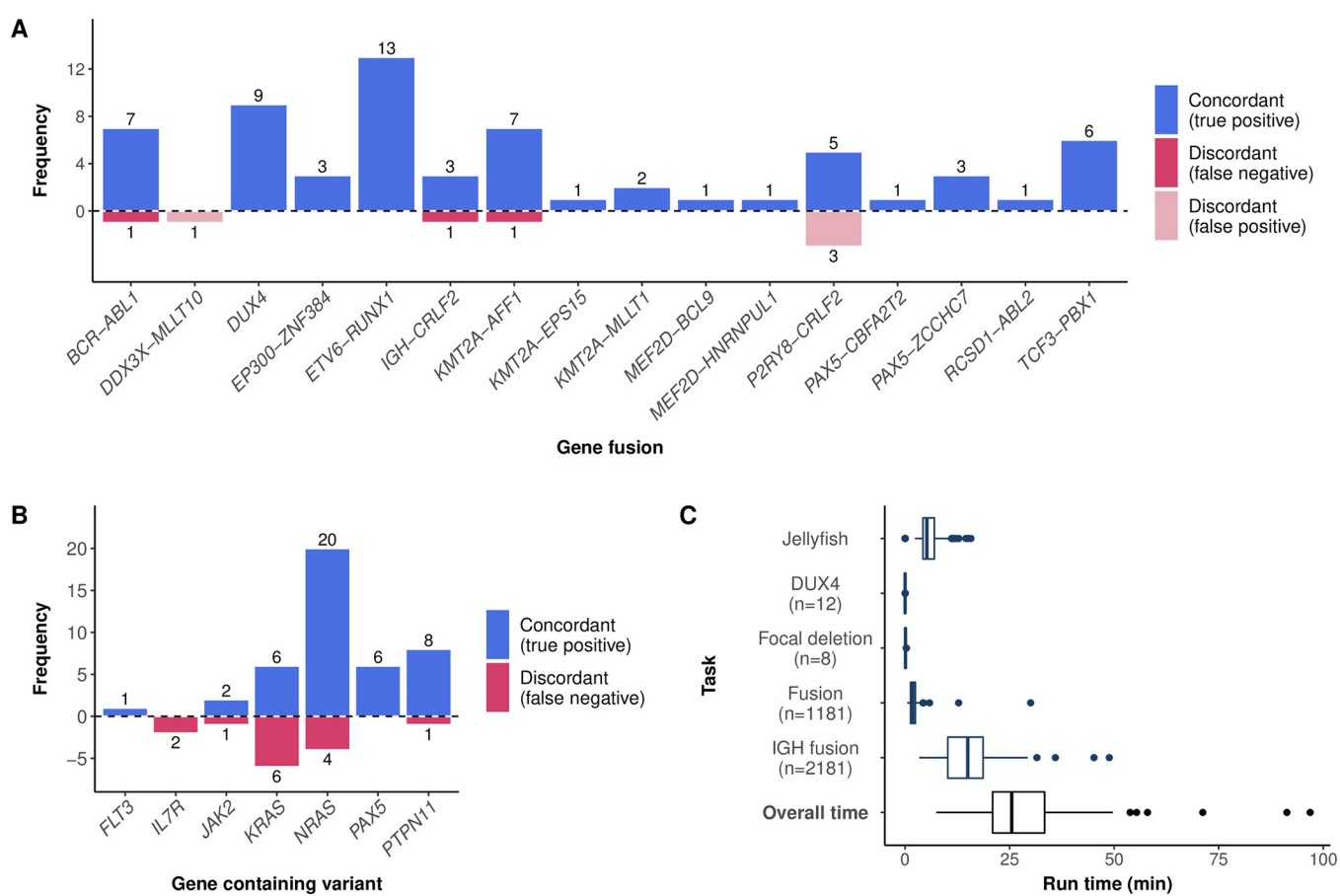

**Fig 4. RaScALL accurately detects subtype defining genetic variants in a validation cohort of 100 B-ALL patient samples.** Fusions (A) and variants (B) detected by RaScALL and their concordance with clinically reported alterations. The vertical axis indicates the number of detected alterations while the horizontal axis indicates the specific gene fusion (A) or gene in which the variant occurred (B). Concordant results are indicated in blue and discordant indicated in dark red for false negative results and light red for false positive results. (C) Runtimes for RaScALL in minutes. Wall clock time for 100 samples represented as boxplots. The vertical axis indicates the separate run times for indexing of raw data and assessment of variants with different target sets. The Number of targets in each set is indicated.

RaScALL in only 2 cases (S5 Table). Runtimes for this random cohort were longer than reported previously, due to increased read length (100bp PE), sequencing depth (median 106 million reads), and an increase in the number of target sequences. Increasing the number of non-*IGH* fusion targets from 39 to 1181 resulted in an average increase of 143 seconds for non-*IGH* gene fusion detection. Overall median runtime was 22.6 minutes, attributed primarily to the assessment of *IGH* gene fusions (Fig 4C). Runtimes exceeding 50 minutes were reported for 3 samples with sequencing depth greater than 140 million PE reads, and all with *IGH-DUX4* fusions expected to result in high-level expression of the *IGH* locus.

## Discussion

RNA-seq produces a comprehensive overview of patient disease, allowing clinicians to accurately assign risk and design personalised treatment plans of chemotherapy with or without targeted therapies. Consequently, several researchers have endorsed the use of this technology as a front-line diagnostic tool in future clinical trials [5–7,68]. However, successful integration of genomic data in a clinical setting requires bioinformatics tools that can rapidly and accurately identify clinically relevant variants [7,22]. Large cohort studies utilising RNA-seq have

identified multiple recurrently observed and prognostically significant genomic alterations for which rapid targeted variant analysis can be performed, reducing the reliance on computationally intensive *de novo* variant detection methods that limit clinical translation speed. This study aimed to develop a bioinformatic tool utilising a target variant detection approach for rapid identification and reporting of clinically relevant genomic alterations in ALL.

Gene fusions represent the most common driver mutations in patients with ALL, and many are associated with a specific risk profile [1–3]. Loss of function mutations of *IKZF1* and *PAX5* are less prevalent, affecting fewer than 1% of paediatric B-ALL patients, but becoming more common in adolescent and young adult (AYA) and adult cohorts with *PAX5* p.P80R detected in 3–4% of patients [5]. RaScALL enables accurate and simultaneous detection of both mutation types, in greater than 95% of patient samples that carry the mutation, within minutes. False-negative results were reported in a small proportion of cases due to either uncommon exon-exon breakpoints not represented in the target set, or the presence of a patient-specific polymorphism in the first or final k-mer of the target sequence. These limitations can be mitigated by relying on public population-scale genetic variation databases, such as 1000 genomes [69] or gnomAD [70], that have identified rare and common variants across a diverse set of global populations. Careful curation of available variant data is recommended when generating potential target sequences.

Gene fusions involving the cytokine receptors *CRLF2* and *EPOR* with *IGH/IGK* occur in approximately 6–14% of B-ALL diagnoses for older children, adolescents and adults and are associated with high-risk disease [22]. However, few of the currently available fusion calling algorithms can reliably detect these alterations due to difficulty mapping to IG loci [23,25]. In this study, we successfully identified *IGH-CRLF2* and *IGH-EPOR* alterations in 95 and 83% of tested cases respectively, demonstrating similar sensitivity to FusionCatcher [30], one of the few fusion calling algorithms capable of detecting *IGH* gene fusions.

*DUX4r* are also frequently missed by alignment-based fusion calling algorithms [23] in addition to being cryptic to standard cytogenetic analysis and fluorescent *in situ* hybridisation [71]. *DUX4* expression is a reliable indicator of *DUX4*r [13,72], however, the presence of additional *DUX4* pseudo-gene copies in the reference genome, showing an almost identical DNA sequence to the annotated *DUX4* gene, complicates the accurate determination of *DUX4* expression by alignment [23,72]. Direct detection of *DUX4* expression across the first 700b of the *DUX4* transcript using RaScALL correlated directly with samples identified as *DUX4*r by gene expression profiling, enabling more sensitive detection of *DUX4*r than alignment-based tools. Given that patients with this ALL subtype typically demonstrate an excellent prognosis, early detection of patients with this lesion can assist in clinical decisions regarding treatment intensity [73].

In addition to reporting subtype defining genomic alterations, RaScALL can detect cooperating missense mutations and focal gene deletions, ensuring rapid reporting of multiple clinically relevant genomic alterations required for appropriate prognostication of patient disease. However, detection of focal gene deletions is limited to intragenic deletions resulting in aberrant transcript isoforms. Whole gene deletions and deletions of exon 1 resulting in loss of allelic transcription cannot be identified from RNA-seq data. For these reasons, DNA-based assays for copy number variant analysis are still recommended where patient material is available. Furthermore, detection of missense variants via pseudo-alignment is sensitive to sequencing coverage, with lowly expressed variants including *JAK2* being missed in a proportion of cases and analysis being limited to a small number of targeted variants. RaScALL requires a minimum of 5 reads spanning the target region to contain the altered allele for reliable detection of sequence variants (i.e., 25X coverage across region of interest for subclonal variants in 20% of sample). Follow-up analysis utilising more sensitive alignment-based *de*

*novo* small variant detection tools (e.g., GATK HaplotypeCaller) is therefore recommended for the identification of less common or lowly expressed sequence variants.

A key advantage of RaScALL is the speed with which it can identify clinically relevant genomic alterations for patient prognostication. Most *de novo* variant detection tools depend upon the alignment of reads, a time and memory-consuming process. Consistent with previous benchmarking studies [23,27] we confirmed mean run times 0.5 hours for the memory intensive fusion calling algorithm *arriba* increasing to 15 hours for the low-memory tool *SOAPfuse*. Sequence variant detection by GATK similarly requires several hours for completion of all steps in the GATK best practices pipeline [74]. In contrast, RaScALL can identify gene fusions and sequence variants of interest in under 30 minutes, with a moderate memory allocation (5-20GB depending on length and depth of sequencing). When implemented within a clinical sequencing laboratory using a hypbrid or commercial cloud computing environment, higher speed and lower memory utilisation translates to lower overall computing costs for bioinformatic analysis [35].

*De novo* variant detection tools can generate false-positive predictions due to sequencing artefacts and alignment issues [23,27,75]. In the case of gene fusions, read-through transcripts and *cis*-splicing can also result in a low number of chimeric transcripts without underlying chromosomal alterations [76]. In our study, this was observed in the validation cohort with the *P2RY8-CRLF2* gene fusion, highlighting the need for caution regarding fusion sequences detected by RaScALL with very low coverage. Likewise, target sequences containing homopolymer stretches can result in false-positive detection of InDels due to sequencing error (e.g. *ILR7* p.S252ins). Importantly, by limiting the search space to known alterations and requiring an exact match to the expected breakpoint sequence we greatly reduce the false positive rate compared to alignment-based variant detection tools. This eliminates the need for consensus between multiple variant detection tools [23,27] or extensive filtering and manual data curation [25], which limit the speed of clinical translation.

Targeted variant detection approaches cannot detect the presence of novel disease-causing variants or identify the full range of variants that may potentially contribute to leukaemic progression. Subtype defining lesions, currently detectable with RaScALL represent 50–70% of paediatric and 60–85% of adult B-ALL samples [2,5,77]. Fewer target sequences are available for detection of subtype defining alterations in T-cell ALL (T-ALL) which represent 15–25% of ALL cases [78]. This is in part due to the greater variability in sequence variants observed [79], the reduced availability of T-ALL samples for RNA-seq analysis and a greater ambiguity around the prognostic significance of T-ALL associated lesions [78]. Given the highly heterogeneous nature of ALL, new genomic alterations of clinical relevance will continue to be identified [7,22,79], while ongoing research will continue to refine the associated prognostic impact of these lesions. As such the repertoire of ALL specific alterations for detection from RNA-seq will grow and change over time. To accommodate this, we have provided command line scripts for generating additional target sequences representing canonical gene fusions, SNVs and InDels to allow for rapid expansion of variants for analysis with RaScALL.

## Conclusion

RNA-seq is being increasingly utilised in clinical trials for ALL as it enables more refined sub-classification of patient samples than standard diagnostic tests. Effective implementation of this technology is however dependent on bioinformatic analysis that allows for accurate and reliable detection of clinically relevant genomic alterations in a clinically relevant time frame. In this setting, where the presence or absence of a finite list of known genomic alterations is needed to determine patient risk, a targeted variant detection approach can be highly

beneficial. RaScALL provides comparable detection of clinically relevant ALL gene fusions and co-occurring mutations and superior detection of *DUX4*r than alignment-based *de novo* variant calling tools, without the need for extensive filtering and manual curation of detected alterations. Runtimes are significantly shorter than most alignment-based tools available for *de novo* gene fusion and sequence variant detection, and the lower memory requirements mean that analysis can be easily scaled across large computational resources. Using the custom target generation tool additional target sequences can be easily generated, allowing users to expand or adapt their analysis to suit reporting guidelines. RaScALL provides a fast, light-weight tool to quickly screen RNA-seq data for clinically relevant alterations, enabling rapid reporting of patient-specific ALL variants to assist with prognostication and treatment allocation.

## Materials and methods

### Ethics statement

Ethical approval for sample collection and analysis was granted by the following Human Research Ethics Committees (HRECs) within South Australia; the Central Adelaide Local Health Network (CALHN) Human Research Ethics Committee and the Women's and Children's Health Network (WCHN) Human Research Ethics Committee. Approval for the collection of samples outside of South Australia was granted under the National Mutual Acceptance Scheme. Informed written consent was obtained from patients, or in the case of children, their legal parent/guardian.

### RaScALL overview

A k-mer count table (k = 31) is generated from patient RNA-seq data using *jellyfish* v2.2.6 [33]. Variant detection is then performed with *km* [28] using the *jellyfish* count table and target sequences as input. Targets are separated according to the type of variant for assessment (Fusion, IGH Fusion, DUX4, SNV, and focal deletion) and *km* find_mutation is performed separately for each variant type, with the output written to a text file. A custom R script then filters and annotates variants generating a single output file containing detected clinically relevant alterations. RaScALL was run on a 32 thread 128 GB RAM HPC node at the South Australian Health and Medical Research Institute (SAHMRI). Code and additional scripts for running RaScALL are available from the GitHub repository (https://github.com/j-rehn/RaScALL).

### Generation of target sequences for variant detection within the study cohort

Target sequences applied to the study cohort (target set A) were designed as previously described [28] for gene fusions involving exon-exon breakpoints and single nucleotide variants, limiting this list to recurrently observed genomic alteration noted in the literature as being clinically relevant. Briefly, for gene fusions, the break-point genomic coordinates reported by FusionCatcher [30] for previously analysed patient samples were used to extract nucleotide sequences from UCSC with the getDNA portal and incorporated 31b of sequence from each gene involved in the fusion. Targets for sequence variants were also generated using the UCSC getDNA portal, ensuring that the reference sequence was in-frame with 33b on either side of the amino acid position(s) of interest. To detect transcripts indicative of *IKZF1* deletions, target sequences containing 31b of each exon on either side of the potential deletion breakpoint were created.

Targets for detection of *DUX4* expression were generated using the first 700b of the NCBI Reference Sequences transcripts of DUX4 (NM_001306068.3) and the hg38 alternate locus

containing DUX4 homolog (NT_187650.1). A total of twelve non-overlapping target sequences of 70b length each were generated, excluding regions between 141-210b of the mRNA transcript which was highly homologous to multiple genomic locations within the human reference genome. Targets representing potential breakpoints for *IGH-CRLF2* and *EPOR-IGH* rearrangements were generated by combining 50bp sequence 5' of reported *CRLF2* and *EPOR* breakpoints with sequences from all *IGHJ*, *IGHD* and *IGK* alleles downloaded from IMGT, the international ImMunoGeneTics information system (http://www.imgt.org).

## Generation of target sequences for variant detection within the validation cohort

Target sequences applied to the validation cohort (target set B) were generated by combining target set A with additional target sequences produced with RaScALL's custom target generation tools (https://github.com/j-rehn/RaScALL). Briefly, tab-separated files were compiled containing gene names, RefSeq transcript identifiers and either breakpoint exon numbers (gene fusions) or amino acid position (SNV/InDels) for the additional gene fusions/ sequence variants of interest. These were supplied to the custom_targets.sh script along with the UCSC hg19 human reference genome and associated gtf file, which identifies the genomic locations required for the target sequence and prints this to a separate FASTA file for each fusion/variant. ALL specific gene fusions, single nucleotide variants and focal gene deletions for clinical reporting were identified from the literature, limited to variants that were multiply reported. To reduce the likelihood of false-negatives target regions involved in gene fusions with exon-exon breakpoints were searched in gnomAD [70] to identify single nucleotide polymorphisms (SNP) with a variant allele frequency greater than 0.01. When identified an additional target sequence containing the SNP was manually generated.

## Patient samples in the study cohort

The study cohort consisted of RNA-seq data from 180 Australian ALL patient samples taken either at diagnosis (n = 149) or relapse (n = 31) selected from a larger pool of patient samples referred to the South Australian Health and Medical Research Institute (SAHMRI) for genomic testing (Adelaide, Australia). The cohort included 160 B-ALL, 18 T-ALL and 2 samples with acute bi-phenotypic leukaemia or acute undifferentiated leukaemia, each with an identified driver mutation or associated aneuploidy representing commonly observed clinically relevant genomic alterations. Library preparation for mRNA sequencing was performed using either Truseq Stranded mRNA LT Kit (Illumina, CA, USA) or Universal Plus mRNA-Seq with NuQuant (Tecan, CA, USA) from 400ng of total RNA as per the manufacturer's instructions. Samples were sequenced by either Illumina HiSeq 2000 or NextSeq 500 platforms producing 75b length paired-end (PE) reads with a median read depth of 70M reads.

Fluorescent in situ hybridization (FISH) was performed on a subset of patient samples showing surface expression of CRLF2 by flow-cytometry. *IGH-CRLF2* rearrangements were detected using two separate break-apart probes, a *CRLF2* dual colour probe (Cytocell) and a Vysis IGH dual colour probe (Abbott). Where material was available *IKZF1* deletions were detected by multiplex ligation-dependent probe amplification using two SALSA Reference kits (P335 and P202, MRC-Holland, Amsterdam, Netherlands) according to the manufacturer's instructions and data was analysed using Coffalyser software.

## *De novo* variant detection performed for the study cohort

RNA-seq data was analysed through an in-house bioinformatics pipeline for the detection of gene fusions and single nucleotide variants (SNV). For SNV detection, raw FASTQ data was

aligned to the human reference genome (GRCh37) with STAR aligner (v 2.5.3) [80] using 2-pass mode. Variant calling was performed by *GATK HaplotypeCaller* [29] and annotated with *Annovar* [81]. Only variants with a minimum of 10 reads over the variant position and a VAF > 0.1 were reported and detected variants with an allele frequency < 0.2 were visualised in IGV [82] to confirm they represented true variants and not sequencing artefacts.

Gene fusions were detected utilising *SOAPfuse* v1.26 [31], *JAFFA* v1.09 [32], *fusionCatcher* (fusioncatcher.py 0.99.7c beta) [30] and *arriba* v2.1.0 [23]. To reduce false positives only fusions detected by multiple callers were considered as likely candidates except for fusions involving *IGH* which are frequently only detected by *fusionCatcher* or to a lesser extent *arriba*. For *IGH-CRLF2* fusions all samples in the study cohort had FISH supporting the presence of the rearrangement, while *IGH-EPOR* rearrangements were confirmed by visual inspection of the BAM file. Gene expression profiling was performed to confirm the presence of rearrangements involving *DUX4*. Gene counts were generated from the STAR aligned samples using featureCounts [83] and TMM normalised with edgeR [84]. Hierarchical clustering was then performed using previously published gene lists [72,85,86].

## Application of RaScALL to the validation cohort

To assess the clinical applicability of RaScALL in a diagnostic setting a validation cohort of 100 samples was selected from publicly available data provided by Gu *et al.* 2019 [5] (European Genome-phenome Archive: EGAD00001004461 and EGAD00001004463). Samples were selected at random after excluding those reported as hyperdiploid or hypodiploid. BAM files were converted to FASTQ using *samtools* (v1.3.1) and the resultant FASTQ files were analysed by RaScALL. Gene fusions, SNVs and InDels reported as part of the Gu *et al.* 2019 [5] study for these samples were utilised to evaluate the accuracy of RaScALL for the detection of clinically relevant variants. Methods describing de novo variant detection of genomic alterations for these samples can be found in the original publication (Gu *et al.* 2019 [5]). SNVs detected by RaScALL but not reported as part of the study were visualised in IGV. Runtimes reported include indexing of FASTQ files with *jellyfish* (12 threads) and variant detection by *km*.

## Memory and run time evaluation

Maximum memory usage for *jellyfish* and *km* was estimated as previously described [34]. For indexing of raw RNA-seq data with *jellyfish*, tests were performed on simulated fastq files generated with randomreads.sh (sourceforge.net/projects/bbmap/) from the Human Ensembl GRCh38 reference transcriptome. Memory requirements for variant detection with *km* were assessed utilising the 180 samples from the study cohort with PE read length 75b and mean read depth 70M. Runtime for *jellyfish* was measured using the bash internal variable SECONDS and averaged over three runs. Elapsed time reported by *km* for a given set of targets was extracted from the raw output files.

An additional evaluation of runtime and memory was also performed between RaScALL and the fusion calling algorithms employed for de novo fusion detection in the study cohort (*FusionCatcher*, *SOAPfuse*, *JAFFA* and *arriba*). For this analysis 9 samples from the study cohort, representing high (80+ million), moderate (50–80 million) and low (< 50 million) read depths, were selected for benchmarking (A2638-12600, ADI-0158, AYAI-0004-REL1, AYAII-0063, CH-A1725, CH-A2497, CH-A5243-DIA, CHI-0452-DIA1-BM, CHI-0511-DIA1-PB). Each sample was processed individually by each program on a HPC node with 16CPU and 128GB RAM. The resource allocation for individual jobs was a maximum of 8CPU and 12 threads. Elapsed time and maximum resident set size (Max RSS) for individual jobs was extracted from the slurm workload manager at completion of analysis.

## Supporting information

**S1 Table. Clinical information and reported genetic alterations for patients in the study cohort.**
(XLSX)

**S2 Table. Summary of target sequences utilised by RaScALL for variant detection within the study cohort (target set A).**
(XLSX)

**S3 Table. Run time and peak memory usage for generating k-mer count table with *jellyfish* using simulated RNA-seq data.**
(XLSX)

**S4 Table. Genomic alterations detected by RaScALL for the study cohort and a comparison against detection of these alterations by alignment-based *de novo* detection tools or MLPA.**
(XLSX)

**S5 Table. Summary of target sequences utilised by RaScALL for variant detection within the validation cohort (target set B).**
(XLSX)

**S6 Table. Clinical information and genomic alterations detected by RaScALL for the validation cohort.**
(XLSX)

**S1 Fig. Hierarchical clustering of B-ALL patients in study cohort utilising differentially expressed genes reported for patients with *DUX4*-rearrangement (*DUX4*r).** Hierarchical clustering of CPM normalised expression data using published gene sets by (A) Yeoh et al., 2002 (85) (B) Harvey et al., 2010 (86) and (C) Zhang et al., 2016 (72). Each column represents a single B-ALL patient. Driver gene fusion identified for the sample is indicated at the top of each heatmap and coloured according to the key (*IGH-DUX4* fusion coloured red). Patients identified as *DUX4*r in the cohort are indicated with dark blue.
(TIF)

**S2 Fig. Comparison of overall runtime and maximum memory usage between RaScALL and four alignment-based fusion calling algorithms (*FusionCatcher*, *Arriba*, *JAFFA*, and *SOAPfuse*).** Elapsed time and maximum resident set size (Max RSS) were obtained for 9 patient samples from the study cohort representing high (80+ million), moderate (50–80 million) and low (< 50 million) read depths. Analysis was performed on a HPC Node with 16CPU and 128GB RAM. Resources allocated for each job was a maximum of 8CPU and 12 threads. Elapsed time and Max RSS were obtained from the slurm workload manager at the completion of individual jobs. Boxplot representing (A) elapsed time (minutes) and (B) Max RSS (GB) for each program. Points represent individual samples coloured according to read depth as indicated in the key above.
(TIF)

**S3 Fig. DUX4 target sequences detected in samples from the study cohort.** Number of DUX4 targets detected in samples within the study cohort (A) and the minimum k-mer coverage across DUX4 target sequences (B) for each sample and reported on a $\log_{10}$ scale. Samples from patients reported as *DUX4*r based on hierarchical cluster analysis are shown in red and non-*DUX4*r patients indicated in blue.
(TIF)

**S4 Fig. Upset plot comparing detection rate of 164 clinically relevant gene fusions utilising both individual programs and combinations of RaScALL, *FusionCatcher*, *arriba*, *JAFFA* and *SOAPfuse*.** (A) Intersection of reported driver gene fusions represented as a percentage of total driver gene fusions present in the cohort (n = 164). Each column corresponds to fusion calling algorithm (final 5 columns) or sets of fusion callers (dots connected by lines below the X axis). Only fusions reported by all fusion callers included in the set are considered. The percentage of driver genes detected in each set appears above the column. (B) Union of reported driver gene fusions represented as a percentage of total driver fusions present in the cohort (n = 164). Fusions reported by any of the fusion callers included in the set are considered. The percentage of driver genes detected in each set appears above the column. (C) Graph of total number of driver gene fusions detected by each fusion calling algorithm.
(TIF)

## Acknowledgments

The authors would like to acknowledge the South Australian Genomics Centre (SAGC, SA, AUS) and the ACRF Cellular Imaging and Cytometry Core Facility (SAHMRI, SA, AUS) for sequencing and imaging services. We would like to acknowledge the New South Wales Ministry of Health-funded Luminesce Alliance for personnel support and the lab of Professor Charles Mullighan at St Jude Children's Research Hospital for providing sequencing data for the validation cohort. Finally, we would like to thank the patients, their families and treating clinicians for their participation in this study.

## Author Contributions

**Conceptualization:** Jacqueline Rehn, James Breen.

**Data curation:** Jacqueline Rehn.

**Funding acquisition:** David T Yeung, Deborah L White.

**Investigation:** Susan L Heatley, Barbara J McClure, Laura N Eadie, Caitlin Schutz.

**Methodology:** Jacqueline Rehn, James Breen.

**Project administration:** Mark J Cowley, James Breen.

**Resources:** Deborah L White.

**Software:** Jacqueline Rehn, Chelsea Mayoh.

**Supervision:** James Breen, Deborah L White.

**Validation:** Chelsea Mayoh.

**Visualization:** Jacqueline Rehn.

**Writing – original draft:** Jacqueline Rehn.

**Writing – review & editing:** Chelsea Mayoh, Susan L Heatley, Barbara J McClure, Laura N Eadie, Caitlin Schutz, David T Yeung, Mark J Cowley, James Breen, Deborah L White.

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
