## [Decision Letter · Decision Letter 0]

25 Jul 2022

Dear Dr Breen,

Thank you very much for submitting your Research Article entitled 'RaScALL: Rapid (Ra) screening (Sc) of RNA-seq data for prognostically significant genomic alterations in acute lymphoblastic leukaemia (ALL)' to PLOS Genetics.

The manuscript was fully evaluated at the editorial level and by independent peer reviewers. The reviewers appreciated the attention to an important problem, but raised some substantial concerns about the current manuscript. Based on the reviews, we will not be able to accept this version of the manuscript, but we would be willing to review a much-revised version. We cannot, of course, promise publication at that time.

Should you decide to revise the manuscript for further consideration here, your revisions should address the specific points made by each reviewer especially reviewer # 2. Importantly, please address the overarching concern, regarding "what is the real need in a clinical/diagnostic lab and whether such labs will use such a pipeline".  We will also require a detailed list of your responses to the review comments and a description of the changes you have made in the manuscript.

If you decide to revise the manuscript for further consideration at PLOS Genetics, please aim to resubmit within the next 60 days, unless it will take extra time to address the concerns of the reviewers, in which case we would appreciate an expected resubmission date by email to plosgenetics@plos.org.

[LINK]

We are sorry that we cannot be more positive about your manuscript at this stage. Please do not hesitate to contact us if you have any concerns or questions.

Yours sincerely,

Ronald B Gartenhaus

Academic Editor

PLOS Genetics

Peter McKinnon

Section Editor: Cancer Genetics

PLOS Genetics

Reviewer's Responses to Questions

**Comments to the Authors:**

Reviewer #1: With this manuscript the authors describe a novel computational algorithm to identify genomic alterations that define subtypes of acute lymphoblastic leukemia from RNA sequencing. The approach uses the k-mer based targeted detection tool and known sets of ALL variants and was compared to standard alignment based pipelines. The tool appears to perform relatively well, particularly in identifying difficult to detect variants in the test cohort and a validation cohort. The approach may have some advantages in terms of required computational resources and time.

This is a potentially useful tool that appears to perform well. A disadvantage is that novel fusions may be missed as the algorithm depends on prior knowledge of variant sequences. This disadvantage is mitigated with the ability to add target sequences when they are reported. What is not discussed is how this algorithm might be implemented clinically, particularly in light of regulatory oversight of laboratory generated data that may be used for clinical decision making.

Reviewer #2: In this study, authors introduce a computational pipeline, RaScALL, for inferring genomic aberrations from RNA-sequencing data in ALL patients. RaScALL pipeline allows for screening and inferring gene fusions, SNVs and focal gene deletions with potential prognostic value for ALL patients. Moreover, RaScALL implements the km algorithm that conducts a pseudo-alignment rather than a direct mapping of raw RNA-seq reads to a reference genome. Authors argue that this approach is more efficient in terms of memory and computational time as it reduces the search space.

The workflow of RaScALL includes implementation of four fusion calling algorithms (FusionCatcher, SOAPfuse, JAFFA and Arriba), and the consensus of at least two of the algorithms was required for gene fusion identification (except for IGH rearrangements, IGHr).

Authors applied RaScALL in a patient cohort of 180 subjects that represent common prognostic genomic lesions in B-ALL (160), T-ALL (18) and mixed lineage leukaemia, MLL (2). They also validated RaScALL inferences via the data analysis of 100 publicly available datasets samples.

The results show that RaScALL successfully identified gene fusions in both cohorts for a fraction of computational time. Their pipeline can accommodate new reports of gene fusions and de novo variants.

I find this paper of great-quality work that makes a contribution to cancer research. Authors present a new computational tool that can benefit the clinicians for disease diagnosis in ALL patients. However, the first big question is what is the real need in a clinical/diagnostic lab and whether such labs will use such a pipeline. Let’s not forget that RNA-Seq is not used routinely for diagnostics on ALL. It is used infrequently, for example in cases that are suspected Ph-like and in most laboratories, samples are sent out for RNA-Seq to reference laboratories. Most places are probably not even sending out such samples. Apart from this general comment, there are several issues that need to be addressed:

• The T-ALL cohort is small and key mutations (NOTCH1, FBXW7, PTEN etc) and not ever represented. The authors should expand the cohort and add such mutations and also ad additional events (MYC duplications, NOTCH rearrangements etc).

• How easy is to modify RaScALL? Add mutations, rearrangements etc? How that would affect the analysis time and flow?

• What is the cost of running RaScALL per sample?

• Lines 132-134: Add the citation of the km algorithm.

• Lines 156-161: There are some inconsistencies here when describing the study cohort. E.g., You have merged the ALL, MLL patients for the data analysis as I can see in the Methods section and Figure 1. But this is not clear here. You should also explain here that there was a validation cohort, different ALL subtypes etc

• Figure1: It would be interesting to see how the identification of genetic fusions change in ALL patients in different disease stages (patients at Diagnosis vs Relapse disease stages). Are there any differences? Maybe consider to add the time point here? Also, you should include here the results from the study and validation cohorts together.

• Line 169: You need to explain why IGHr were excluded. It seems that this is explained later as “regions with highly variable breakpoint locations” (Line 197). But you should elaborate more on this, and add a citation.

• Lines 171-172 and Lines 397-398: You mention that your study cohort contained patient of high sequencing coverage (mean 70M PE reads). How is the performance of RaScALL affected by the read coverage? Provide minimum read coverage or suggestions for the user to deal with such patient cases.

• Line 189: are the identified pathogenic SNVs first-time reported here (thanks to RaScALL) or are these based on known SNVs?

• Lines 250-251: This information is partially stated as part of the beginning in the Results section (Lines 156-160); consider merging these two parts.

• Line 304: To add citation about MLPA analysis.

• Line 326: Caption should be more self-explanatory. E.g., explain a little bit what MLPA analysis is. Are the results here related to the detection of IKZF1 intragenic deletions by using RaScALL and comparing them to the MLPA analysis?

• Line 385, Figure 4: You should show the overall performance, accuracy and run time of RaScALL pipeline comparing to the other methods (FusionCatcher, SOAPfuse, JAFFA and Arriba), and how these change as you get concordance from 2, 3 or all 4 methods.

• Lines 430-431: Maybe, it would be useful for researchers/clinicians to get a prognostic risk score or a label that would indicate ALL subtype and/or disease stage based on the identification of the genetic lesions, that would help patient stratification and indicate progression of the disease.

**Have all data underlying the figures and results presented in the manuscript been provided?**

Reviewer #1: Yes

Reviewer #2: Yes

PLOS authors have the option to publish the peer review history of their article (what does this mean?). If published, this will include your full peer review and any attached files.

Reviewer #1: No

Reviewer #2: No

---

## [Editor Report · Decision Letter 1]

22 Sep 2022

Dear Dr Breen,

We are pleased to inform you that your manuscript entitled "RaScALL: Rapid (Ra) screening (Sc) of RNA-seq data for prognostically significant genomic alterations in acute lymphoblastic leukaemia (ALL)" has been editorially accepted for publication in PLOS Genetics. Congratulations!

Yours sincerely,

Ronald B Gartenhaus

Academic Editor

PLOS Genetics

Peter McKinnon

Section Editor

PLOS Genetics

Comments from the reviewers (if applicable):

**Data Deposition**

http://datadryad.org/submit?journalID=pgenetics&manu=PGENETICS-D-22-00730R1

**Press Queries**

---

## [Editor Report · Acceptance letter]

12 Oct 2022

PGENETICS-D-22-00730R1 

RaScALL: Rapid (Ra) screening (Sc) of RNA-seq data for prognostically significant genomic alterations in acute lymphoblastic leukaemia (ALL) 

Dear Dr Breen, 

We are pleased to inform you that your manuscript entitled "RaScALL: Rapid (Ra) screening (Sc) of RNA-seq data for prognostically significant genomic alterations in acute lymphoblastic leukaemia (ALL)" has been formally accepted for publication in PLOS Genetics! Your manuscript is now with our production department and you will be notified of the publication date in due course.

With kind regards,

Anita Estes

PLOS Genetics

On behalf of:
